# Soft Sensor Modeling Method for the Marine Lysozyme Fermentation Process Based on ISOA-GPR Weighted Ensemble Learning

**DOI:** 10.3390/s23229119

**Published:** 2023-11-11

**Authors:** Na Lu, Bo Wang, Xianglin Zhu

**Affiliations:** Key Laboratory of Agricultural Measurement and Control Technology and Equipment for Mechanical Industrial Facilities, School of Electrical and Information Engineering, Jiangsu University, Zhenjiang 212013, China; 2212107042@stmail.ujs.edu.cn (N.L.); zxl4390@126.com (X.Z.)

**Keywords:** marine lysozyme, seagull optimization algorithm, Gaussian process regression, soft sensor, grayscale correlation analysis

## Abstract

Due to the highly nonlinear, multi-stage, and time-varying characteristics of the marine lysozyme fermentation process, the global soft sensor models established using traditional single modeling methods cannot describe the dynamic characteristics of the entire fermentation process. Therefore, this study proposes a weighted ensemble learning soft sensor modeling method based on an improved seagull optimization algorithm (ISOA) and Gaussian process regression (GPR). First, an improved density peak clustering algorithm (ADPC) was used to divide the sample dataset into multiple local sample subsets. Second, an improved seagull optimization algorithm was used to optimize and transform the Gaussian process regression model, and a sub-prediction model was established. Finally, the fusion strategy was determined according to the connectivity between the test samples and local sample subsets. The proposed soft sensor model was applied to the prediction of key biochemical parameters of the marine lysozyme fermentation process. The simulation results show that the proposed soft sensor model can effectively predict the key biochemical parameters with relatively small prediction errors in the case of limited training data. According to the results, this model can be expanded to the soft sensor prediction applications in general nonlinear systems.

## 1. Introduction

Marine lysozyme has the characteristics of low action temperature, wide pH range, strong activity at room temperature, and moderate decrease in activity as temperature decreases [1,2]. It brings new energy and opportunities to industries such as cleaning, medicine, environmental protection, and food processing [3,4]. Therefore, it is necessary to dynamically regulate and optimize the marine lysozyme fermentation process in real-time to maximize its production efficiency and product quality. However, the marine lysozyme fermentation process is a multivariate, time-varying, and complex nonlinear process. Due to practical process technology and cost considerations, key biochemical parameters that directly reflect the fermentation quality, such as cell concentration, substrate concentration, and relative enzyme activity, can only be roughly estimated through offline sampling and analysis. This process not only affects the operator’s ability to make accurate decisions regarding real-time response status, but also limits the implementation of the best control method. Therefore, it is urgent to find a method that can predict the key biochemical parameters of marine lysozyme fermentation accurately in real time.

Soft sensor technology is an effective method to solve the above problems [5,6,7,8,9]. Hua et al. [10] proposed a new hybrid soft sensor model based on RF-IHHO-LSTM (random forest-improved Harris hawks optimization-long short-term memory) for the penicillin fermentation process. The simulation results show that the established soft sensor model has higher measurement accuracy and better effect, which can meet the practical requirements of the project. Wang et al. [11] constructed a multi-output least squares support vector machine (MLSSVM) regressor model to solve the problem of multi-input and multi-output for l-lysine. They also introduced the Improved Cuckoo Search (ICS) algorithm to optimize the essential parameters of the model. Finally, the hybrid ICS-MLSSVM soft-sensor model was used to predict lysine key parameters online. The simulation results show that the proposed regression model could accurately predict key biochemical parameters. Tokuyama et al. [12] developed a novel soft sensor model for estimating substrates, bacterial cells, and the concentration of target products in commercial fermenters. The results suggest that the machine learning-based soft sensor model could represent a novel monitoring system for digital transformation in the biotechnology process field. Wang et al. [13] used an artificial neural network model to develop soft sensors to monitor lipolytic yeast’s microbial lipid fermentation process. The results show that this model offers the possibility of monitoring stem cell weight, glucose concentration, and lipids online with high accuracy. Sun et al. [14] developed a SOM-LSSVM (SOM, self-organizing feature map; LSSVM, least squares support vector machine) global modeling method for predicting the fermentation potency of CTC (CTC, potency of chlortetracycline). Field experiments show that the method could obtain more accurate potency prediction values.

Although the above modeling methods can meet the basic requirements of key biochemical parameters for online prediction, how can the prediction accuracy of the model be further improved? Due to the characteristics of multiple operating conditions, strong nonlinearity, and high uncertainty in the biological fermentation process, sample data under different operating conditions often have significant differences. Therefore, traditional global soft sensor models constructed using a single soft sensor modeling method cannot accurately describe the dynamic characteristics of the entire fermentation process, resulting in low prediction accuracy and poor generalization ability of the model. This makes it challenging for global soft sensor models to describe the multi-stage nature of the fermentation process, so these applications cannot guarantee prediction accuracy in the global scope. Some scholars suggest applying ensemble learning to this issue. Ensemble learning is an advanced machine learning method that combines different fusion strategies with basic models to achieve more accurate predictions. The basic idea is that even if a weak base model obtains an incorrect prediction, other weak base models can still correct this error. Usually, ensemble learning has a more substantial generalization capability than the base model. Due to its flexible adaptability, ensemble learning has been successfully applied in various fields. Shen et al. [15] proposed a new method based on stochastic programming to realize a quality-related monitoring scheme for batch processing of multiple output modes through ensemble learning. Wang et al. [16] established a prediction model for rumen fermentation parameters in dairy cows by a stacked ensemble learning method and in vitro technique. The comparison results show that the stacking ensemble learning method had better prediction results. Shen et al. [17] proposed a multivariate trajectory based on an ensemble punctual learning strategy to realize a batch quality prediction scheme for the problem of batch diversity. The literature indicates that the modeling method based on ensemble learning theory frameworks can improve the accuracy and generalization ability of a single global soft sensor models. It can be considered for the prediction of key biochemical parameters in the fermentation process.

Considering the excellent characteristics of ensemble learning and the nonlinear, multivariate, and multi-stage features of the marine lysozyme fermentation process, this paper proposes a weighted ensemble learning soft sensor modeling method based on an improved seagull optimization algorithm and Gaussian process regression [18,19,20,21] (ISOA-GPR). The structure of ISOA-GPR weighted ensemble learning is shown in Figure 1. Firstly, an improved density peak clustering algorithm (ADPC) is used to partition subsets of local samples and generate ISOA-GPR sub-prediction models. Then, the improved grayscale correlation algorithm is used to extract the centroids of each local sample subset. The centroid is weighted by information entropy to generate a weighted “centroid” that more accurately represents the local sample subset features. Finally, a fusion strategy based on weighted improved grayscale correlation algorithm is proposed by selecting the sub-prediction models that highly correlated with the test samples. Applying the constructed soft sensor model to the problem of predicting bacterium concentration, substrate concentration, and relative enzyme activity in marine lysozyme fermentation, the simulation results show that compared with the single global soft sensor model based on ISOA-GPR, the prediction error and volatility of this model are smaller.

## 2. Theoretical Analysis

### 2.1. Data Subsets Construction Method

In response to the data distribution of marine lysozyme fermentation, this paper proposes an improved density peak clustering algorithm (ADPC) to partition the data subsets. This density-dependent classification method evaluates the similarity between samples of data and can be used to cluster datasets of arbitrary shapes. Density peak clustering (DPC) is a typical method based on density clustering [22]. This algorithm requires each data point on which classification relies to have two eigenvalues: local density $\rho_{i}$ and relative distance $\delta_{i}$. It assumes that the cluster center has a more significant local density and a larger relative distance from other cluster centers than other data points.

Relative distance refers to the minimum distance between a sample point and other points of higher density. For the sample set R, the local density $\rho_{i}$ of data $x_{i}$ is
(1)$$\rho_{i}=\sum_{i\neq j}\exp\left(-{\left(\frac{dist_{ij}}{dist_{c}}\right)}^{2}\right)$$
where $x_{i}$ and $x_{j}$ represent the *i* th and *j* th data points, respectively. $dist_{ij}$ is the distance between data $x_{i}$ and $x_{j}$, and $dist_{c}$ is the truncation distance.

Due to the higher local density and relative distance of the DPC algorithm’s clustering center compared to other data points, a multiplication of the two is used to select the clustering center. If the cluster centers are nearby, it is not easy to accurately select them by continuing to use the above foundation. Therefore, this paper chooses a logarithmic function to emphasize the differences between the clustering center and other data points. The process is shown below.

Define a decision parameter $D_{i}$ that combines local density, relative distance, and logarithmic functions. Then, name the decision parameters $\gamma_{i}$ in descending order and deduce the downward trend according to Equation (3):(2)$$D_{i}=\rho_{i}\times \mathrm{lg}\left(\delta_{i}\right)$$
(3)$$\gamma_{i}^{*}=\frac{\gamma_{i-1}-\gamma_{i}}{\gamma_{i}-\gamma_{i+1}}$$
where $\gamma_{i}$ represents the current γ value and $\gamma_{i-1}$ $\gamma_{i+1}$ represent the γ values at the preceding and subsequent places, respectively. According to the distribution of downward trend, select the data points with the highest downward trend and those before it as the clustering centers. The flowchart of the ADPC algorithm is shown in Figure 2.

Applying this method to actual marine lysozyme data samples, the descending distribution of decision parameters and the downward trend of parameters $\gamma_{i}$ are obtained, as shown in Figure 3 and Figure 4, respectively. From Figure 3, it can be seen that there is often a significant difference in decision parameters between clustering and non-clustered centers. Except for the first few data points, the decision parameters of the other data points have little fluctuation. They are not suitable for being selected as clustering centers.

### 2.2. Sub-Prediction Model Construction

Considering the obvious nonlinear characteristics of the marine lysozyme fermentation process, small data sample size, and difficulty in offline extraction, etc., this paper selects the Gaussian process regression method, which is good at predicting complex nonlinear outputs using small sample data, to establish a prediction model for the marine lysozyme fermentation process [23]. For Gaussian process regression models, the selection of hyperparameters has a significant impact on the accuracy of the prediction mode precision. Traditional parameter selection methods rely on empirical and trial and error, making it difficult to ensure regression accuracy and calculation speed. In order to generate a sub-prediction model with better performance, this paper utilizes the improved seagull optimization algorithm (ISOA) to optimize and adjust hyperparameters online.

### 2.3. Improved Seagull Optimization Algorithm

SOA is an intelligent algorithm that simulates the behavior of seagull flocks in nature. This algorithm solves the spatial iterative optimization problem by simulating the seasonally varying long-range migratory behavior and spiral attack behavior of seagulls [24]. Compared with other optimization algorithms, the algorithmic structure of the SOA method is simpler, more stable, and more adaptable. It only has one modifiable parameter in the actual optimization process.

In traditional seagull optimization algorithms, the weight parameters linearly decrease with the increase in iteration times. Although the algorithm executes quickly, each iteration can easily lead to a decrease in population diversity. It is easy to have the problem of weak global search ability in the early stage of the algorithm and poor local mining ability in the late stage of the algorithm. So, this paper proposes a nonlinear change in the weight parameter updating strategy. The specific expression is as follows:(4)$$A=-f_{C}\times \tan\left[\frac{t}{Max_{iteration}}\times \frac{\pi }{4}-\frac{\pi }{4}\right]$$
where t is the current number of iterations, $Max_{iteration}$ is the maximum number of iterations, and $f_{C}$ is a constant whose initial value is set to 2.

The improved weight parameters first decrease rapidly as the number of iterations increases, and then gradually and slowly decrease. In the early iterations of the improved seagull optimization algorithm, the weight parameters suddenly decrease to maintain population diversity, which can also improve its global search capability. In the later stages of execution, the weight parameters gradually decrease, thereby improving the local search ability, which also ensures that the algorithm is not easily trapped in a local optimum. Therefore, using an improved seagull optimization algorithm to optimize hyperparameters will undoubtedly result in a more accurate soft sensor mode. The pseudocode of ISOA algorithm is given in Algorithm 1.
**Algorithm** **1:** Improved Seagull Optimization Algorithm (ISOA) **Input**: seagull population Pos **Output**: optimal search agent bestPos
1: Initialize the parameters maxIter, population and func
 /*here func represents the fitness function*/
2: **procedure** ISOA

3: set *f*_c_ ← 2

4: set *u* ← 1

5: set *v* ← 1
 /*Initialize Pos*/
6: Pos ← Init(population,dim,lb,ub)
 /*Initialize the Pos of each seagull agent using Init function*/ /*here dim represents the dimension of the given problem*/ /*here ub, lb represent the upper and lower bounds*/
7: fitness ← func(Pos)

8: [sortfitness, index] ← sort(fitness)

9: bestfitness ← sortfitness(1)

10: bestPos ← Pos(index(1), :)

11: **for** *t* ← 1 to maxIter **do**
/*Migration behavior*/
12:  *A* ← −*f*_c_ × tan(t/maxIter×pi/4−pi/4)

13:  *Cs* ← *A*.×Pos

14:  *B* ← 2. × *A*^2^.×rand(population,1)

15:  *Ms* ← *B*. × (bestPos−Pos)

16:  *Ds* ← abdz(Cs + Ms)
/*Attacking behavior*/
17:  theta ← rand(population,1). × 2.×pi

18:  *r* ← *u*. × exp(theta. × v)

19:  *x* ← *r*. × cos(theta)

20:  *y* ← *r*. × sin(theta)

21:  *z* ← *r*. × theta

22:  Pos ← Ds. × x. × y. × z +Pos
/*Update optimal search agent*/
23:  **for** i ← 1 to population **do**

24:     fitness(i) ← func(Pos(i), :)

25:     if (fitness(i) < bestfitness) then

26:     bestfitness ← fitness(i)

27:     bestPos ← Pos(i, :)

28:  **end for**

29: **end for**

30: return bestPos

30: **end procedure**


### 2.4. Sub-Prediction Model Selection and Fusion Strategy

Fusion strategy is an important component of ensemble learning, and a good fusion strategy is the key to demonstrating its superior performance. This paper adopts a fusion strategy based on improved grayscale correlation to determine the weights of sub-prediction models. This fusion strategy finds the weighted “centroid” of every local sample subset, which can best represent the feature of the entire data subset. Then, a better sub-prediction model is selected based on the correlation between the test sample and the weighted “centroid”. The correlation coefficient between the test samples and the weighted “centroid” is analyzed by using the improved grayscale correlation algorithm because the improved grayscale correlation algorithm can more accurately reflect the fluctuation between the marine lysozyme fermentation data sequences. Given a local sample subset of the marine lysozyme fermentation process data $r=\left\{x_{i};i=1,2,\ldots ,n\right\}$, where $x_{i}\in R^{d},n$ is the number of samples in each local sample subset and d is the feature variable’s dimensionality, let the reference sequence be $x_{0}=\left\{x_{0}\left(1\right),x_{0}\left(2\right),\ldots ,x_{0}\left(d\right)\right\}$ and calculate the grayscale correlation coefficient:(5)$$\varsigma_{i}\left(k\right)=\frac{\underset{i}{\min}\underset{k}{\min}\left|\Delta \right|+\rho \underset{i}{\max}\underset{k}{\max}\left|\Delta \right|}{\left|\Delta \right|+\rho \underset{i}{\max}\underset{k}{\max}\left|\Delta \right|},\rho \in [0,1]$$
where $\Delta =\sqrt{{\left(x_{0}\left(k\right)-\bar{x_{0}}\right)}^{2}}-\sqrt{{\left(x_{i}\left(k\right)-\bar{x_{i}}\right)}^{2}},\left(k=1,2,\ldots ,d\right)$ $\bar{x_{i}}=\frac{1}{d}\sum_{k=1}^{d}x_{i}\left(k\right)$, and ρ indicates the resolution coefficient, which is taken as 0.5.

The correlation between the reference and comparison sequences is calculated as follows:(6)$$\varphi_{i}=\frac{1}{d}{\sum \varsigma }_{i}\left(d\right)$$

Let each sample of the local sample subset be the reference sequence and the remaining of that local sample subset samples be comparison sequences. The generated sample correlation matrix is computed as follows:(7)$$\phi =\left(\begin{array}{cccc} 1 & \varphi_{12} & \cdots & \varphi_{1n}\\ \varphi_{21} & 1 & \cdots & \varphi_{2n}\\ \vdots & \vdots & \ddots & \vdots \\ \varphi_{n1} & \varphi_{n2} & \cdots & 1 \end{array}\right)$$

The sample with the strongest correlation with all local sample subsets data is picked as the data set’s initial center of mass. Its correlation coefficients with other samples in that local sample subset are reported to generate the correlation coefficient matrix:(8)$$\psi =\left(\begin{array}{cccc} \zeta_{1}\left(1\right) & \zeta_{1}\left(2\right) & \cdots & \zeta_{1}\left(k\right)\\ \zeta_{2}\left(1\right) & \zeta_{2}\left(2\right) & \cdots & \zeta_{2}\left(k\right)\\ \vdots & \vdots & \ddots & \vdots \\ \zeta_{n}\left(1\right) & \zeta_{n}\left(2\right) & \cdots & \zeta_{n}\left(k\right) \end{array}\right)$$

In this paper, the information entropy is used to characterize the degree of variation for each feature variable under the correlation coefficient matrix in a weighted manner. This process assigns objective weights to the feature variables and generates a weighted “centroid” that is more representative of the information in the local sample subset. In general, the lower a feature variable’s information entropy, the larger its degree of variation and the higher its given weight. Conversely, when information entropy increases, the relevance of feature variables decreases, and weights decrease. The characteristic weight of the jnd characteristic variable of the ith sample is calculated as:(9)$$P_{ij}=\frac{\zeta_{i}\left(j\right)}{\Sigma_{i=1}^{n}\zeta_{i}\left(j\right)}$$

The entropy value of the j characteristic variable:(10)ej=−1lnn∑i=1nPij∗lnPij,Pij≠0limPij→0Pij∗lnPij=0,Pij=0

Then, the weights of each characteristic variable in the correlation coefficient matrix are obtained as follows:(11)$$w_{j}=\frac{1-e_{j}}{d-\sum_{i=1}^{d}e_{j}}$$

Assume that the improved density peak clustering method collects a total of m local sample subsets. The initial center of mass of the m subsets is named $Z^{*}$ to obtain the weighted “centroid” $Z_{m}$($Z_{m}=w_{j}Z^{*}$). Then, the correlation set $\omega =\left[\omega_{1},\omega_{2},\omega_{3},\ldots ,\omega_{m}\right]$ is obtained by setting the fermentation test sample $x^{*}$ as the reference sequence and the weighted “ centroid “ of m local sample subsets as the comparison sequence. Generally speaking, $\omega^{*}$ is picked as the critical correlation coefficient, which means that the ensemble learning will retain the ISOA-GPR sub-prediction models corresponding to correlation coefficients greater than or equal to $\omega^{*}$. Its corresponding fermentation process sub-prediction model result is $y_{pre}=\left[y_{pre1},y_{pre2},y_{pre3},\ldots ,y_{pre\eta }\right],\eta \in \left[1,m\right]$, so the final prediction result of grayscale correlation weighted ensemble learning is:(12)$$y_{prediction}=\frac{\omega_{1}}{\sum \omega_{\eta }}y_{pre1}+\frac{\omega_{2}}{\sum \omega_{\eta }}y_{pre2}+\frac{\omega_{3}}{\sum \omega_{\eta }}y_{pre3}+\cdots +\frac{\omega_{\eta }}{\sum \omega_{\eta }}y_{pre\eta }$$

## 3. Modeling Process

The modeling flowchart of the ISOA-GPR Weighted Ensemble Learning is shown in Figure 5. To better illustrate the process of building the soft sensor model in this paper, the modeling process is described as follows.

Step 1: Obtain data on the marine lysozyme fermentation process through experiments, including major environmental parameters and key biochemical parameters (bacterium concentration, substrate concentration, relative enzyme activity). The improved density peak clustering algorithm is utilized to divide local sample subsets ($R=\left\{r_{1},r_{2},\ldots ,r_{m}\right\}$) as well as to calculate the weighted “centroid” $Z_{m}$($Z_{m}=w_{j}Z^{*}$) for each local sample subset.

Step 2: Calculate the correlations degree between various environmental parameters and key biochemical parameters. Select environmental parameters with correlations greater than 0.7 as auxiliary variables to build ISOA-GPR sub-prediction models.

Step 3: Send the test sample ($x^{*}$), calculate its grayscale correlation coefficient with each weighted “centroid” ($\omega_{i}\left(i=1,2,\ldots ,m\right)$), and select the ISOA-GPR model ($\omega_{i}>0.7$) as sub-prediction models. Determine the output weights of each sub-prediction model and output predictions according to Equation (12).

## 4. Simulation Results and Analysis

In order to validate the effectiveness of the proposed soft sensor modeling method, this paper simulates the data from the marine lysozyme fermentation process. Taking the marine lysozyme fermentation process as the object, the culture strain was S-12-86, and the fermenter model was A103-500L. The Yellow Sea Fisheries Research Institute of the Chinese Academy of Fisheries Sciences gave the marine lysozyme fermentation method, and the Jiangsu University fermentation control system platform provided the navigational lysozyme fermentation data.

In this paper, bacterium concentration (*X*), substrate concentration (*S*), and relative enzyme activity (*P*) were taken as the most key biochemical parameters in marine lysozyme fermentation. An improved grayscale correlation algorithm was used to filter auxiliary variables and extract data from 15 fermentation batches. The first 12 batches, a total of 720 data points, were used as training samples and the last 3 batches, with a total of 180 data points, were used as test samples. These measurements’ values were used for training simulations based on a single global ISOA-GPR model, an ISOA-BP-weighted ensemble learning soft sensor model and an ISOA-GPR-weighted ensemble learning soft sensor model. The simulation results are depicted in Figure 6, Figure 7, Figure 8, Figure 9, Figure 10 and Figure 11. To show that the ISOA-GPR weighted ensemble learning soft sensor model performs better, root mean square and maximum absolute errors compare how well the three models can predict. The results are displayed in Table 1.
(13)$$e_{RMSE}=\sqrt{\frac{1}{m}\sum_{i=1}^{m}{\left(y^{\left(i\right)}-{\hat{y}}^{\left(i\right)}\right)}^{2}}$$
(14)$$e_{MAE}=\max\left|y^{\left(i\right)}-{\hat{y}}^{\left(i\right)}\right|$$
where $y^{\left(i\right)}$ represents the values of all actual key biochemical parameters (bacterial concentration, substrate concentration, and relative enzyme activity) for the tested samples. ${\hat{y}}^{\left(i\right)}$ represents the values of all predicted key biochemical l parameters (bacterial concentration, substrate concentration, and relative enzyme activity) for the tested samples.

This paper established two sets of comparison experiments with ISOA-GPR single global, ISOA-GPR weighted ensemble learning and ISOA-GPR weighted ensemble learning; and ISOA-BP weighted ensemble learning. The predicted curves of key biochemical parameters (bacterium concentration, substrate concentration, and relative enzyme activity) for the marine lysozyme fermentation process were derived using each of the three models, as shown in Figure 6, Figure 8 and Figure 10. In order to clearly compare the prediction accuracy, the absolute error curves of the three key biochemical parameters predicted (bacterial concentration, substrate concentration, and relative enzyme activity) are shown in Figure 7, Figure 9 and Figure 11, respectively.

From the above figures and table analysis the following can be seen.

(1) It can be concluded from Figure 6 that all three models can predict the bacterium concentration well, and that the curve trend of the prediction model and the actual data are basically the same. It can be obtained in Figure 7 that, except for some individual data points (e.g., the 15th data point), the ISOA-GPR weighted ensemble learning model has the lowest prediction error and the ISOA-GPR single global model has the highest prediction error. Compared with the ISOA-GPR single global model and the ISOA-BP weighted ensemble learning model, the ISOA-GPR weighted ensemble learning model predicts a 68.6% and 48.2% lower root mean square error, respectively.

(2) When combined with Figure 8 and Figure 9, the ISOA-GPR single global model prediction curve fluctuates the most near the turn of the matrix concentration curve. After the fluctuation smoothed out, the prediction error is the smallest, while the ISOA-BP weighted ensemble learning model prediction error instead becomes larger and larger. The ISOA-GPR weighted ensemble learning model prediction error is basically controlled in the range of [−1, 1] for the whole range. Compared with the ISOA-GPR single global model and the ISOA-BP weighted ensemble learning model, the ISOA-GPR weighted ensemble learning model predicts a 52.8% and 39% lower root mean square error, respectively.

(3) When combined with Figure 10 and Figure 11, the ISOA-GPR single global model has the largest prediction error in the online prediction of relative enzyme activity and the error range could reach up to 8%. Compared with the ISOA-GPR single global model and the ISOA-BP weighted ensemble learning model, the ISOA-GPR weighted ensemble learning model predicts a 77.7% and 55.3% lower root mean square error, respectively.

(4) Comparing the prediction results of the single global ISOA-GPR model and the weighted ensemble learning ISOA-GPR model for bacterial concentration, substrate concentration and relative enzyme activity, it is intuitively found that the prediction results based on the weighted ensemble learning ISOA-GPR model are significantly less volatile than those based on the single global ISOA-GPR model. The root mean square prediction errors are 68.6%, 52.8% and 77.7% lower, respectively, allowing for better monitoring of the actual values of key biochemical parameters.

(5) Comparing the prediction results of the single global ISOA-GPR model and the weighted ensemble learning ISOA-BP model for the three key biochemical parameters, it is found that the weighted ensemble learning ISOA-BP model increased the prediction accuracy to a certain extent, but not as much as the weighted ensemble learning ISOA-GPR model did. In the late stage of substrate concentration prediction, the prediction error of the weighted ensemble learning ISOA-BP model increased significantly. So, the weighted ensemble learning ISOA-GPR model has some advantages in the prediction process of the three biochemical parameters.

## 5. Conclusions

In order to solve the problem in which a single global soft sensor model of the marine lysozyme fermentation process has poor reliability and cannot guarantee the global prediction accuracy, this paper presents an ISOA-GPR weighted ensemble learning soft sensor model. Firstly, an improved density peak clustering algorithm is used to adaptively partition the fermentation process data. Then, a soft sensor sub-prediction model for the fermentation process is established using an improved seagull optimization algorithm and the Gaussian process regression (ISOA-GPR). Finally, an improved grayscale correlation algorithm is used to extract the entropy weighted “centroid” of the subset, and the comprehensive output of the sub prediction model is filtered. The simulation results show that the root mean square errors of the model are 0.5333, 0.8103, and 0.8439, respectively, which can be used to predict key biochemical parameters based on less training data. Its prediction error is small and sufficient to meet the demand for online measurement of key biochemical parameters of marine lysozyme.

Compared with the traditional single global soft sensor model, the method proposed in this paper is more in line with the actual situation of marine lysozyme fermentation at the theoretical level. This not only enables online prediction of the key biochemical parameters in the marine lysozyme fermentation process, but also effectively improves its online prediction and tracking ability, solving the problem of poor prediction accuracy of traditional single global soft sensor models. At the same time, it provides new solutions for other complex nonlinear online prediction industries. Under ideal experimental conditions, the conclusion of this paper are obtained. However, in the actual fermentation process, fermentation conditions are prone to sudden changes. Minor differences in fermentation conditions will affect the predicted results. Therefore, the next stage of research is to solve the problem of how to apply this method to practical complex fermentation processes.

## Figures and Tables

**Figure 1 sensors-23-09119-f001:**
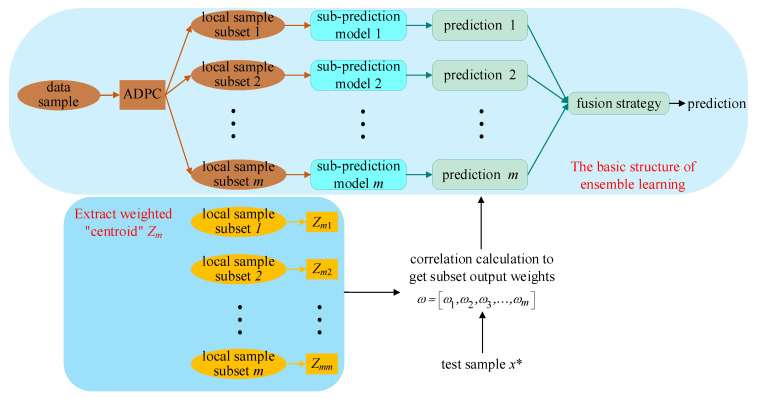
The structure of ISOA-GPR weighted ensemble learning for marine lysozyme fermentation process.

**Figure 2 sensors-23-09119-f002:**
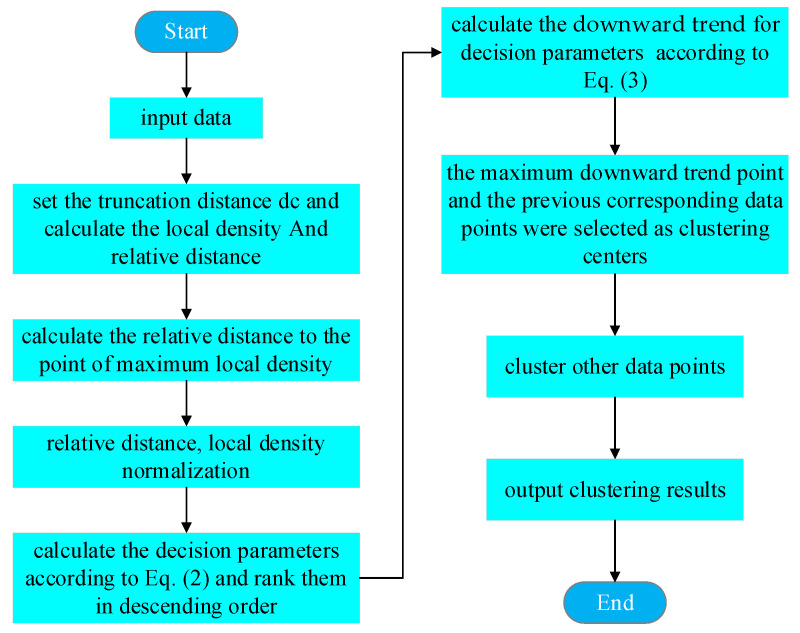
Flowchart of the ADPC algorithm.

**Figure 3 sensors-23-09119-f003:**
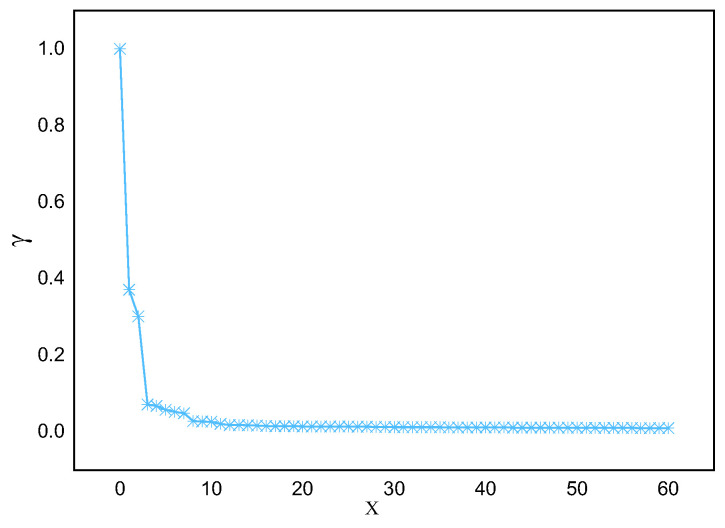
Decision parameter D distribution in descending order.

**Figure 4 sensors-23-09119-f004:**
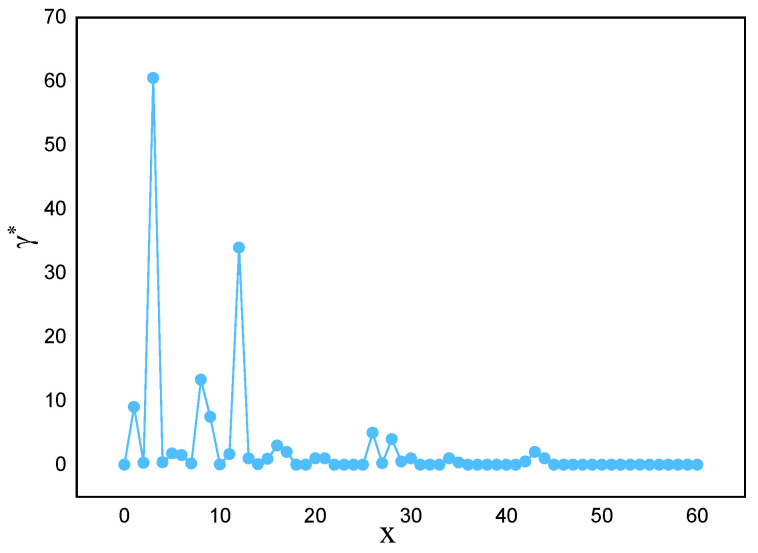
The tendency for decision parameters $\gamma_{i}$ to decrease.

**Figure 5 sensors-23-09119-f005:**
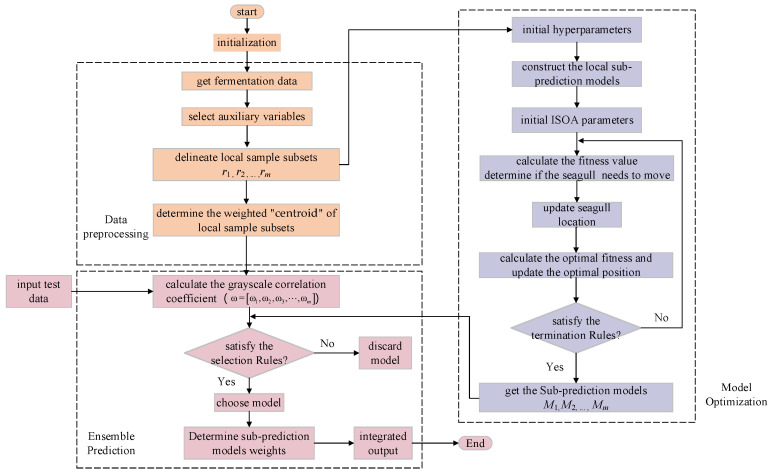
Soft sensor model specific flowchart.

**Figure 6 sensors-23-09119-f006:**
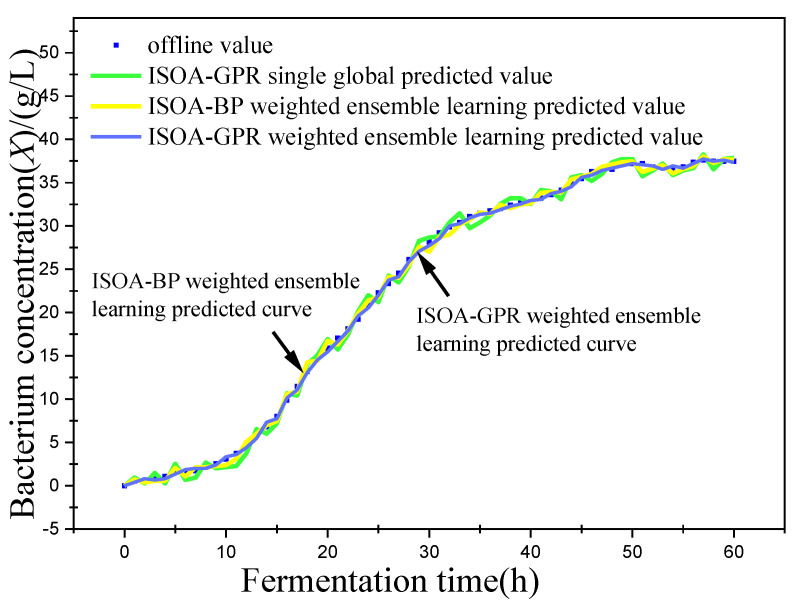
Predicted curve of bacterium concentration.

**Figure 7 sensors-23-09119-f007:**
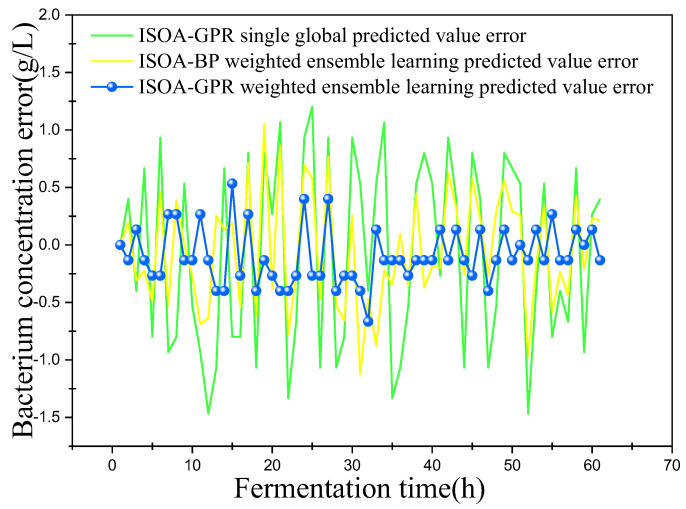
Error variation curve of bacterium concentration.

**Figure 8 sensors-23-09119-f008:**
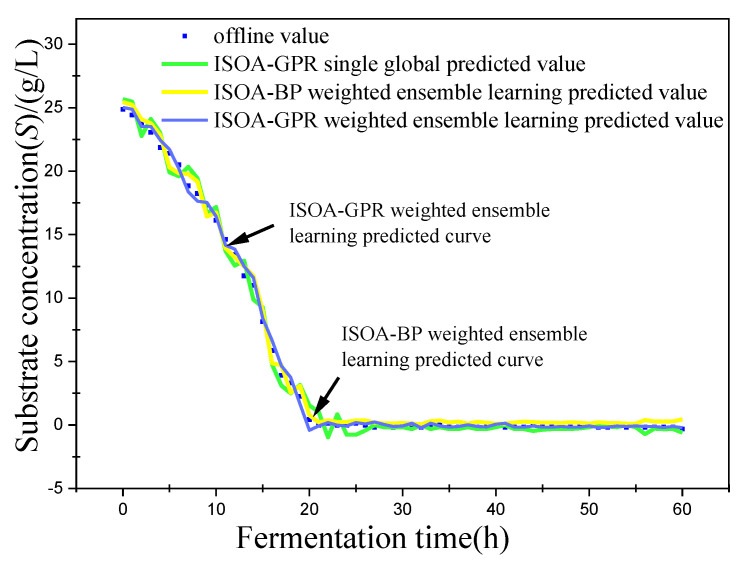
Predicted curve of substrate concentration.

**Figure 9 sensors-23-09119-f009:**
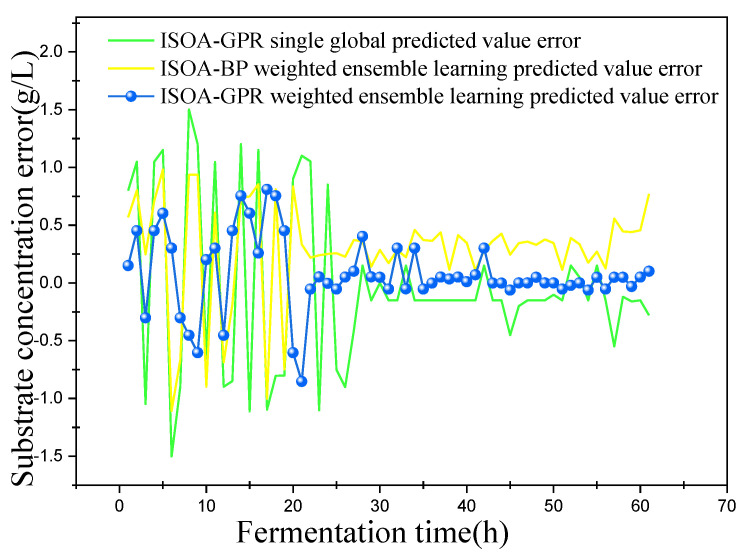
Error variation curve of substrate concentration.

**Figure 10 sensors-23-09119-f010:**
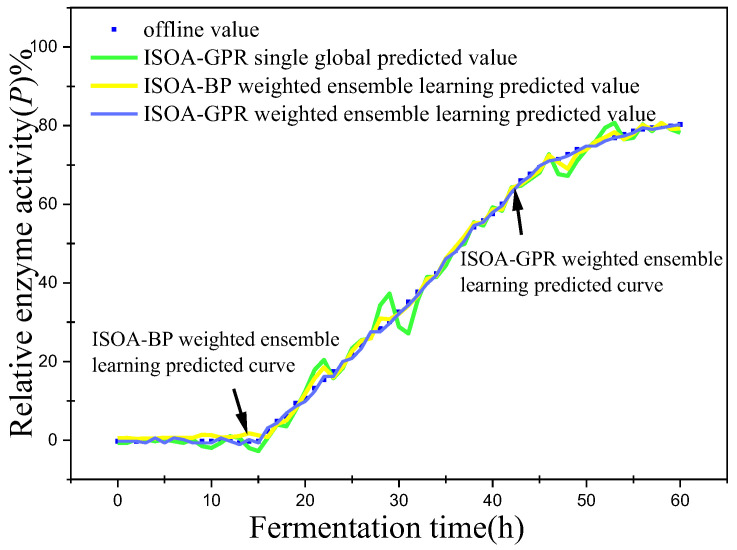
Predicted curve of relative enzyme activity.

**Figure 11 sensors-23-09119-f011:**
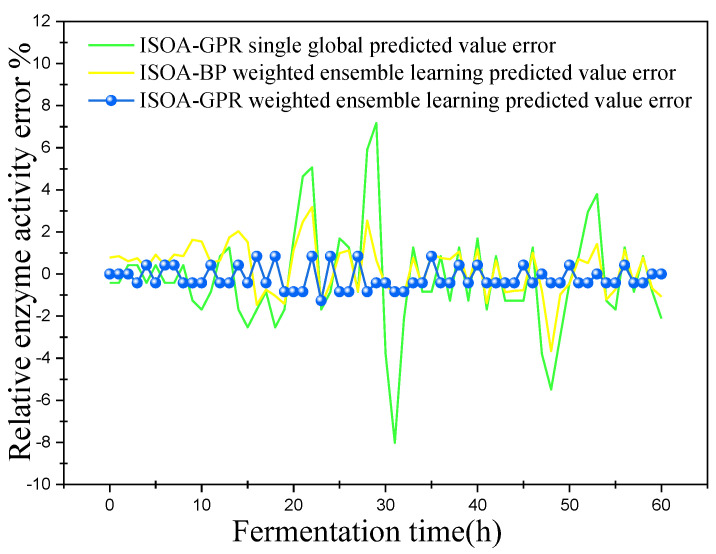
Error variation curve of relative enzyme activity.

**Table 1 sensors-23-09119-t001:** Comparison of the errors of the two modeling methods.

Modeling Method	*e_MAE_*			*e_RMSE_*		
	*X*	*S*	*P*	*X*	*S*	*P*
Single global ISOA-GPR model	1.2	1.5017	7.1730	0.8153	0.6946	2.4651
Weighted ensemble learning ISOA-BP model	1.05	0.9820	3.1801	0.4943	0.5381	1.2316
Weighted ensemble learning ISOA-GPR model	0.5333	0.8103	0.8439	0.2561	0.3281	0.5509

## Data Availability

The dataset in this paper is unavailable because it involves privacy.

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
