# Peer review of "Soft Sensor Modeling Method for the Marine Lysozyme Fermentation Process Based on ISOA-GPR Weighted Ensemble Learning"

_sensors, 2023, doi:10.3390/s23229119_

Round 1
Reviewer 1 Report (Previous Reviewer 3)
Comments and Suggestions for Authors
I don't have any further comments.
Comments on the Quality of English LanguageExtensive editing of English language required.
Author Response
Please see the attachment.

Reviewer 2 Report (Previous Reviewer 1)
Comments and Suggestions for Authors
The paper titled "A Soft Sensor Model based on ISOA-GPR Weighted 3 Ensemble Learning for Marine Lysozyme Fermentation Process" showcases the diligent work of the writers, and may be deemed suitable for potential publication upon implementing significant revisions.
Could you kindly provide an explanation of the concepts CTC and SOM LSSVM? There are other terms inside these concepts that have not been adequately defined.
In the second paragraph of the introduction, the authors fail to provide an explanation of the specific characteristics and limitations of the classic soft sensor in terms of its accuracy, as compared to the newer version. In a similar vein, it is pertinent to discuss the improvements that have been included in the ICS-MLSSVM framework, which render it superior to conventional soft sensors. In summary, the field of literature need renovation.
It is advisable to verify the alignment of the arrows preceding the process of combination as depicted in Figure 1.
The term "gamma" refers to a decision parameter. Can you help confirm this?
The flow charts depicted in Figures 2 and 5 do not adhere to the standard format. I kindly request that you redraw them using appropriate shapes.
Section 3 will provide an explanation of the concept of the weighted centre of mass, denoted as Zm or wi.
In order to enhance reader comprehension, it is recommended that symbols be incorporated into Figure 6 to 9 for the ISOA-GPR and ISOABP lines.
It is recommended to include a concise summary of Figure 6-9 within the theoretical framework figure by figure.
Round 2
Reviewer 2 Report (Previous Reviewer 1)
Comments and Suggestions for Authors
The paper still needs minor changes, as although the authors mentioned they made the changes, they are still not amended, so I am rewriting my suggestions:
1) It is advisable to verify the alignment of the arrows preceding the process of combination, as depicted in Figure 1.
2) The flow charts depicted in Figures 2 and 5 do not adhere to the standard format. I kindly request that you redraw them using appropriate shapes.
3) Axis bars in figures are missing.
4) It is recommended to include a concise summary of Figure 6-9 within the theoretical framework figure by figure, which means adding details according to each figure. Because figures have a lot of details and usually new researchers cannot extract them, we have to explain it to them.
Author Response
Please see the attachment.

This manuscript is a resubmission of an earlier submission. The following is a list of the peer review reports and author responses from that submission.
Round 1
Reviewer 1 Report
Comments and Suggestions for Authors
In the article " A soft sensor model based on ISOA-GPR weighted ensemble learning for marine lysozyme fermentation process " authors are proposing software based study to solve the above problem, this study innovatively proposed a soft sensor modeling method based on an improved seagull optimization algorithm (ISOA) combined with Gaussian process regression (GPR) weighted. This paper needs major revision and then can be considered for further publication:
1. The work is simulation based and looks quite ideal, it is suggested to add some details regarding its practical applications.
2. Recheck all the graphs. Some don’t have axis ticks, some are inside, some are outside, they all should follow same style.
3. Avoid multiple declarations for the same symbol, like local density is defined twice in last and second last line above Equation 1.
4. i and j are not explained anywhere, if they mean initial and final value, please add in the theory.
5. Recheck Figure 1 graph and caption, are both same.
6. Figure 1, 4, 5, 6, 7, 8, 9 need some explanation in the theory.
7. Equation 4 brackets are missing, due to which the equation is confusing.
8. Headings first letter should be capitalized.
9. If some headings are numbered, then the same pattern should be followed.
10.xi is considered 0.5 in which range of k.
11. Font and size between Eq 8 and Eq 9 should be revised.
Author Response
请参阅附件。

Reviewer 2 Report
Comments and Suggestions for Authors
This review critically evaluates the paper proposing an innovative soft sensor modeling approach for the complex marine lysozyme fermentation process. The paper highlights the inherent challenges of modeling this process due to its nonlinear, multi-stage, and strongly time-varying nature, particularly in the context of achieving model stability and prediction accuracy through conventional single global soft sensor models. The study introduces a novel solution by combining the improved seagull optimization algorithm (ISOA) with Gaussian process regression (GPR) weighted ensemble learning. The proposed method involves several stages, including data partitioning using the improved density peak clustering algorithm (ADPC), optimization of the Gaussian process regression model through ISOA for sub-prediction model creation, and a dynamic fusion strategy based on test sample-local subset connections. Simulation results illustrate the success of the proposed soft sensor model in accurately predicting key biochemical parameters of the marine lysozyme fermentation process while utilizing fewer training data. The paper also posits the potential extension of this method to general nonlinear systems' soft sensor modeling.
Strengths:
-
Relevance of Problem: The paper effectively addresses the challenges of modeling a complex marine lysozyme fermentation process, which is characterized by its nonlinear, multi-stage, and time-varying behavior. This problem's significance is well-established within the context of process engineering.
-
Innovative Methodology: The study introduces a novel approach by merging ISOA with GPR weighted ensemble learning. This innovative fusion acknowledges the complexity of the problem and leverages advanced techniques to enhance modeling accuracy.
-
Localized Modeling Strategy: The employment of ADPC to divide sample data into local subsets demonstrates a practical understanding of nonlinear systems, allowing the model to capture nuanced behavior across varying conditions.
-
Algorithmic Enhancement: The introduction of an improved version of ISOA for optimizing the Gaussian process regression model adds an original algorithmic contribution to the study, enhancing its robustness.
-
Fusion Strategy: The dynamic fusion strategy based on the connection between test samples and local subsets showcases a thoughtful consideration of adaptability and enhances the model's performance.
Suggestions for Improvement:
-
Algorithmic Clarity: Providing detailed pseudocode or algorithmic descriptions for ISOA, ADPC, and the fusion strategy determination would facilitate readers' comprehension and replication.
-
Theoretical Context: Expanding on the theoretical rationale behind selecting ISOA and GPR weighted ensemble learning, and their synergy, would strengthen the paper's theoretical foundation.
-
Validation: While simulation results are promising, including validation against real-world data from the marine lysozyme fermentation process would bolster the paper's practical credibility.
-
Comparative Analysis: Comparing the proposed method against conventional single global soft sensor models and other relevant techniques would provide clearer insights into its advantages and limitations.
Conclusion: The paper under review contributes significantly to the field of soft sensor modeling by addressing the complexities of the marine lysozyme fermentation process. The strengths of the paper lie in its problem relevance, innovative methodology, localized modeling strategy, and dynamic fusion mechanism. Addressing the suggested improvements could enhance the paper's clarity and theoretical grounding. Overall, the study's contributions have the potential to advance soft sensor modeling in complex nonlinear systems and process engineering, with implications for broader applications.
Comments on the Quality of English LanguageExtensive editing of English language required
Reviewer 3 Report
Comments and Suggestions for Authors
1- The authors should ask the help of native English speaking proof reader,
because there are too many typo and linguistic mistakes that should be
fixed.
2- Abstract to modify: the abstract should contain Objectives,
Methods/Analysis for ISOA-GPR weighted ensemble learning for marine lysozyme fermentation, and Novelty /Improvement. It is suggested to
present the abstract in one 200 words paragraph that mainly on real-time weed identification and spot spraying applications.
3- The introduction is poorly written and it does not properly refer to
previously published studies. The authors need to carefully review the
published literature, identify the gaps in the literature, and propose their
approach to fill the gap of the research for ISOA-GPR weighted ensemble learning for marine lysozyme fermentation.
4- Literature review is not enough. It is important to add some recent work
(2018-2023) to the literature review. At least 8 new references should be
added to article for ISOA-GPR weighted ensemble learning for marine lysozyme fermentation.
5- We prefer if you use the third person singular, instead of the first
person singular or plural (e.g. "we").
6- An editable version of all figures should be added.
7- Some references are not international and valid. Authors should use
from international resources. Maximum 4 local references are allowed on the areas of for ISOA-GPR weighted ensemble learning for marine lysozyme fermentation.
8- Much more explanations and interpretations should be added for the
result, which are not enough for ISOA-GPR weighted ensemble learning for marine lysozyme fermentation.
9- It is suggested to compare the results of the present study with previous
studies and analyze their results completely for ISOA-GPR weighted ensemble learning for marine lysozyme fermentation.
10- It is suggested to organize the conclusion section much better for ISOA-GPR weighted ensemble learning for marine lysozyme fermentation. This section should be presented in one or two 250-300 word paragraphs.
11- The citation of references in the text doesn’t follow the format
requested by the journal. References must be numbered in order for real-time weed identification and spot spraying applications.
12- The manuscript does not follow the format requested by the Journal; it
should be improved.
The authors must revised major and re submitted
Round 2
Reviewer 1 Report
Comments and Suggestions for Authors
The paper lacks a convincing theoretical framework, which is necessary to be considered for publication.
Thank you
Reviewer 2 Report
Comments and Suggestions for Authors
The author has revised the paper according to my comments. I recommend it for publication in your reputed journal.